# Wound Healing Effect of Supercritical Carbon Dioxide *Datura metel* L. Leaves Extracts: An In Vitro Study of Anti-Inflammation, Cell Migration, MMP-2 Inhibition, and the Modulation of the Sonic Hedgehog Pathway in Human Fibroblasts

**DOI:** 10.3390/plants12132546

**Published:** 2023-07-04

**Authors:** Warintorn Ruksiriwanich, Pichchapa Linsaenkart, Anurak Muangsanguan, Korawan Sringarm, Pensak Jantrawut, Chaiwat Arjin, Sarana Rose Sommano, Yuthana Phimolsiripol, Francisco J. Barba

**Affiliations:** 1Department of Pharmaceutical Sciences, Faculty of Pharmacy, Chiang Mai University, Chiang Mai 50200, Thailand; pichchapa_li@cmu.ac.th (P.L.); anurak_m@cmu.ac.th (A.M.); pensak.j@cmu.ac.th (P.J.); 2Cluster of Valorization and Bio-Green Transformation for Translation Research Innovation of Raw Materials and Products, Chiang Mai University, Chiang Mai 50200, Thailand; korawan.s@cmu.ac.th (K.S.); sarana.s@cmu.ac.th (S.R.S.); 3Cluster of Agro Bio-Circular-Green Industry, Faculty of Agro-Industry, Chiang Mai University, Chiang Mai 50100, Thailand; yuthana.p@cmu.ac.th; 4Department of Animal and Aquatic Sciences, Faculty of Agriculture, Chiang Mai University, Chiang Mai 50200, Thailand; chaiwat.arjin@cmu.ac.th; 5School of Agro-Industry, Faculty of Agro-Industry, Chiang Mai University, Chiang Mai 50100, Thailand; 6Department of Preventive Medicine and Public Health, Food Science, Toxicology and Forensic Medicine, Faculty of Pharmacy, University of Valencia, Burjassot, 46100 Valencia, Spain; francisco.barba@uv.es

**Keywords:** *Datura metel*, cell migration, sonic hedgehog pathway, stimulation of collagen synthesis, supercritical carbon dioxide extraction, wound healing

## Abstract

*Datura metel* L. (thorn apple) has been used in Thai folk wisdom for wound care. In this study, we chose supercritical carbon dioxide extraction (scCO_2_) to develop crude extraction from the leaves of the thorn apple. The phytochemical profiles were observed using liquid chromatography–quadrupole time-of-flight mass spectrometry (LC-QTOF-MS). The biological activities of *D. metel* were performed through antioxidant assays, anti-inflammation based on the Griess reaction, the migration assay, the expression of matrix metalloproteinase-2 (MMP-2), and regulatory genes in fibroblasts. Dm1 and Dm2 extracts were obtained from scCO_2_ procedures at different pressures of 300 and 500 bar, respectively. Bioactive compounds, including farnesyl acetone, schisanhenol B, and loliolide, were identified in both extracts. The antioxidant properties of both *D. metel* extracts were comparable to those of l-ascorbic acid in hydrogen peroxide-induced fibroblasts with no significant difference. Additionally, Dm1 and Dm2 significantly inhibited the nitrite production levels of 1.23 ± 0.19 and 1.52 ± 0.05 μM, respectively, against the lipopolysaccharide-treated group (3.82 ± 0.39 μM). Interestingly, Dm1 obviously demonstrated the percentage of wound closure with 58.46 ± 7.61 and 82.62 ± 6.66% after 36 and 48 h of treatment, which were comparable to the commercial deproteinized dialysate from the calf blood extract. Moreover, both extracts were comparable to l-ascorbic acid treatment in their ability to suppress the expression of MMP-2: an enzyme that breaks down collagen. The gene expressions of *SHH*, *SMO*, and *GLI1* that control the sonic hedgehog pathway were also clearly upregulated by Dm1. Consequently, the scCO_2_ technique could be applied in *D. metel* extraction and contribute to potentially effective wound closure.

## 1. Introduction

*Datura metel* L. var. fastuosa, or *Datura fastuosa*, is a medicinal herb belonging to the genus *Datura* sp. in the Solanaceae family. Various parts of the thorn apple have wide-ranging health benefits as medicine for pains, respiratory, or skin diseases [1]. In Thai traditional medicine, the utility of *D. metel* leaves in infectious skin and cutaneous lesions, including abscesses, tinea infections, insect bites, and chronic wounds, is well-known [2]. Furthermore, bioactive substances related to alkaloids, terpenoids, or saturated fatty acids have been largely presented in thorn apple leaves [3,4,5]. The leaves of *D. metel* potentially manifest antioxidant, anti-inflammatory, anti-bacterial, and anti-fungal effects [1,4]. In addition, previous studies have confirmed that the leaves of *D. metel* could be applied to cutaneous wounds, diabetic ulcers, and psoriasis skin [4,6]. However, all extracts from previous studies were obtained from maceration [4,5] and reflux extraction [3,6].

The key mechanisms of cutaneous wound healing consists of four steps: homeostasis, inflammation, proliferation, and remodeling. The first stage of wound repair is preventing excessive blood loss. Initially, platelets are immediately recruited into the damaged blood vessels and form blood clots with fibrin fibers and extracellular matrix (ECM) proteins [7,8]. Moreover, the growth factors secreted from platelets play crucial roles in immune cell infiltration into injury sites and the proliferation of keratinocytes and fibroblasts [8,9]. These cellular activities occur in subsequent steps of the inflammatory and proliferative stages.

The presence of immune cells and modulating inflammatory cytokines are involved in the response toward bacterial invasion. The recruitment of monocytes, one of the immune cells, turns them into differentiated pro-inflammatory macrophages, namely M1, which are able to phagocytose pathogens and cell debris, as well as promote an inflammatory response [10,11]. On the other hand, delayed wound repairs are associated with an overexpression of pro-inflammatory cytokines, proteolytic enzymes production, and a high abundance of reactive oxygen species (ROS) [7]. Previous data have found that prolonged inflammation contributes to the poor prognosis of chronic wounds [12,13]. Eventually, M2 macrophages are subsequently activated to repress inflammation and recover the adjacent edges of wounds via accelerating keratinocytes, fibroblasts, and endothelial cell migration [11]. These consequences lead to the next stage in wound healing: enclosing skin lesions.

Proliferation, re-epithelization, and angiogenesis are the primary processes for the skin barrier recovery. Keratinocyte and fibroblast cells are predominantly induced to proliferate and migrate to the unhealed edges of wounds [10]. Previous studies have reported that the activation of Wnt/β-catenin and sonic hedgehog signaling pathways can result not only in the migration of resident skin cells, mainly keratinocytes and dermal fibroblasts [14,15], but also in the proliferation of hair shafts [16,17,18]. Afterwards, blood vessels are developed in the wound bed and supply the necessary oxygen and nutrients [10]. It has been demonstrated that growth factors, such as the vascular endothelial growth factor (VEGF), produced from keratinocytes and fibroblasts, are essential for the angiogenesis of endothelial cells [8,19].

Finally, tissue remodeling starts to occur with the emigration of immune cells and the deposition of collagen fibers. This process usually takes over 1–2 years [8,10]. Indeed, the production of collagen is mainly mediated by fibroblast cells. Additionally, the proteolytic activity of matrix metalloproteinases (MMPs) is required for the degradation of ECM components before the formation of collagen networks [7,20]. Conversely, excess MMPs activity may be the cause of poor wound recovery [21].

At present, there are no scientific reports that establish the green extraction of *D. metel* leaves as a wound healing treatment. In this study, we employed the supercritical carbon dioxide (scCO_2_) technique in the extraction of *D. metel* leaves. The active constituents will be characterized using a mass spectrometer. Then, the wound healing activity of the sample will be performed throughout the phases of normal wound healing, including the determination of antioxidant and anti-inflammatory activities that could benefit in the early stage of wound healing: the inflammatory phase. Together with the proliferative and remodeling steps of cutaneous regeneration after wounding, the migration capacity of fibroblasts, MMP-2 activity, and gene expression of Wnt/β-catenin, VEGF, and sonic hedgehog signaling pathways will be studied.

## 2. Results

### 2.1. Supercritical Carbon Dioxide (scCO_2_) Extraction Conditions and Extraction Yields of Datura metel

The effect of extraction pressure was correlated with the extraction yield, as shown in Table 1. The yield of Dm2 extract was 3.0 ± 0.1% *w*/*w* based on dry material, followed by Dm1 extract (2.7 ± 0.2 % *w*/*w*). The extracts obtained from the scCO_2_ extraction were sticky, green-colored semisolids. Afterwards, both extracts were further identified for bioactive compounds.

### 2.2. Phytochemical Constituents from Datura metel Extracts

In the present study, our library could identify bioactive compounds from Dm1 and Dm2 extracts by liquid chromatography–quadrupole time-of-flight mass spectrometry (LC-QTOF-MS). The chromatogram of extracts was illustrated in Figure 1. The highest relative contents of the 12 phytocompositions are shown in Table 2. The most predominant compounds found in both extracts were alkaloids and farnesyl acetone. According to recent studies, farnesyl acetone seems to be responsible for wound healing effects [22,23,24].

### 2.3. Cell Viability

In order to evaluate the biological effect within the cells, each extract at a concentration ranging from 0.00195 to 0.0625 mg/mL was tested using the SRB colorimetric assay. The Dm1 and Dm2 extracts were non-toxic (>80% cell viability) in human fibroblasts and RAW 264.7 cells at concentrations of 0.00195 and 0.0039 mg/mL after 72 h of incubation, as shown in Figure 2. The higher concentrations contributed to cytotoxic effects on the cells. Thus, the maximum non-toxic concentration for each cell line would be used in subsequent experiments.

### 2.4. Antioxidant Properties

#### 2.4.1. Antioxidant Activities against DPPH and ABTS Radicals

The screening of antioxidant properties was determined in terms of the fifty percent inhibitory concentration (IC_50_) against 2,2-diphenyl-1-picrylhydrazyl (DPPH) and 2,2′-azinobis (3-ethylbenzothiazoline-6-sulphonic acid) (ABTS) radicals. The antioxidant properties of leaf extracts were tested at concentrations ranging from 0.039–0.625 mg/mL, as presented in Table 3. The results demonstrated that all extracts provided promising antioxidant potential. Consequently, the intracellular antioxidant capacity would be confirmed through the cell-based assay.

#### 2.4.2. Cellular Antioxidant Activity

The thiobarbituric acid reactive substances (TBARS) assay was used to examine the malondialdehyde production in hydrogen peroxide (H_2_O_2_)-induced human fibroblasts, as shown in Figure 3. The antioxidant capacities of the extracts were compared to the standard scavenger, l-ascorbic acid (0.00195 mg/mL), as a positive control. The TBARS level of H_2_O_2_-activated fibroblasts (118.24 ± 2.50% of control) was significant different from the untreated group (*p <* 0.05). Both Dm1 and Dm2 showed cellular activity with decreased TBARS levels at 93.19 ± 4.42 and 95.29 ± 8.46% of control, with no significant difference to the l-ascorbic acid group (96.77 ± 0.43% of control).

### 2.5. Anti-Inflammatory Activity

The inhibitory effects of *D. metel* extracts on nitric oxide production for anti-inflammatory activity are presented in Figure 4. The positive control for the anti-inflammatory assay was the standard diclofenac sodium (0.0039 mg/mL), non-steroidal anti-inflammatory drug. The level of nitrite was significantly elevated to 3.82 ± 0.39 μM in macrophages activated with lipopolysaccharide (LPS) (*p <* 0.05). Pretreatment with Dm1 and Dm2 could suppress nitrite accumulation to 1.23 ± 0.19 and 1.52 ± 0.05 μM, respectively. On the other hand, diclofenac sodium treatment could notably reduce the nitrite formation (0.50 ± 0.06 μM) than *D. metel* treatment by about 2.50-fold.

### 2.6. Migration of Fibroblasts

Wound closure effects of Dm1 and Dm2 are illustrated in Figure 5. The wound area from cell scratching in all groups decreased in a time-dependent manner. In this study, the highest migration capacity was seen in the treatment of commercial product from calf blood extract for weeping wounds and burns. The wound closure rates after 24, 36, and 48 h of deproteinized dialysate from the calf blood extract treatment were 50.86 ± 6.85, 60.86 ± 5.31, and 88.11 ± 2.97%, respectively. Furthermore, the percentages of wound closure were 36.77 ± 7.09, 58.46 ± 7.61, and 82.62 ± 6.66% in the Dm1 group after 24, 36, and 48 h of treatment. Indeed, there were no significant differences in the migration capacity of fibroblasts at 24, 36, or 48 h between the commercial deproteinized dialysate from the calf blood extract and Dm1 treatment (*p* > 0.05). On the other hand, Dm2 did not assist the wound closure during the first 36 h after scratching. When compared to untreated group, Dm2 had the lowest migration rate, covering 69.52 ± 3.05% of the initial scratch in 48 h (*p* < 0.05).

### 2.7. Expression Levels of Matrix Metalloproteinase Type 2

The protective effects of scCO_2_ extracts on collagen degradation in H_2_O_2_-damaged fibroblast cells were expressed as the inhibition of MMP-2 activity, as shown in Figure 6. The level of active MMP-2 in fibroblast cells was significantly increased to 145.86 ± 15.34% of control after induction with H_2_O_2_ (*p <* 0.05), resulting in the high possibility of collagen degradation. In this study, Dm1 and Dm2 showed remarkable down-regulation of MMP-2 activity (75.92 ± 17.22 and 65.73 ± 18.78% of control) with a comparable level to l-ascorbic acid (76.59 ± 16.98% of control), a cofactor in collagen hydroxylation (*p* > 0.05), which could support the collagen production in the cell remodeling stage of wound healing.

### 2.8. Effects of Datura metel Extracts on Signaling Pathways in Human Fibroblasts

We examined human fibroblasts for the vascular endothelial growth factor-encoding gene (*VEGF*), β-catenin-encoding gene (*CTNNB1*), sonic hedgehog gene (*SHH*), smoothened gene (*SMO*), and glioma-associated oncogene homologue 1 (*GLI1*) levels after *D. metel* treatment, as shown in Figure 7. *VEGF* expression demonstrates regulation via angiogenesis. The expression of *CTNNB1* is related to the Wnt/β-catenin signaling pathway. The results revealed that Dm extracts did not influence the mRNA levels of *VEGF* and *CTNNB1* (Figure 7A,B). Nonetheless, genes associated with the sonic hedgehog pathway, including *SHH*, *SMO*, and *GLI1*, were upregulated after exposure to the Dm1 extract. The expression of *SHH* mRNA was activated by Dm1 (fold change of 1.43 ± 0.05) with no significant difference to the standard purmorphamine (fold change of 1.37 ± 0.09), as represented in Figure 7C. In addition, Dm1 treatment increased the level of *SMO* mRNA with a fold change of 1.51 ± 0.03 (Figure 7D). Remarkably, there were no significant differences among Dm1, the commercial deproteinized dialysate from the calf blood extract (fold change of 1.51 ± 0.07), and the standard purmorphamine (fold change of 1.45 ± 0.16) on *SMO* gene regulation. As shown in Figure 7E, the expression of *GLI1* could be distinctly induced by Dm1 treatment (fold change of 1.75 ± 0.05), compared with the commercial deproteinized dialysate from the calf blood extract treatment (fold change of 1.74 ± 0.06). Accordingly, Dm1 could elevate the sonic hedgehog pathway through the modulation of *SHH*, *SMO*, and *GLI1,* while the commercial deproteinized dialysate from the calf blood extract only upregulated the genes encoded by *SMO* and *GLI1*, which also mediated the sonic hedgehog pathway.

## 3. Discussion

Many previous studies have utilized the conventional methods for *D. metel* extraction, including maceration or reflux extraction with organic solvents. The yields of crude extracts obtained from conventional extraction ranged from 1.4% to 24.1% *w*/*w* [3,4,6]. An inert property of non-polar gas, CO_2_, is used as a removable solvent in an alternative extraction, scCO_2_. The scCO_2_ process is considered safe, environmentally friendly, and known as green technology. This extraction technique requires a co-solvent, e.g., distilled water, ethanol, or methanol, for the extraction of polar or semi-polar substances [25]. Generally, ethanol is the most common solvent in personal care products and easy to evaporate out of the samples [26]. Due to our preliminary study, we tested various extraction methods for *D. metel* extraction, such as maceration, scCO_2_ with water, or ethanol as a co-solvent. When compared to 50% ethanol *D. metel* extracts in our earlier preliminary study, scCO_2_
*D. metel* extracts with 95% ethanol as a co-solvent showed the highest potential wound healing efficacy. As a consequence, Dm1 and Dm2 extracts from the scCO_2_ with 95% ethanol were selected to report the mechanism underlying wound healing properties.

Based on suspect screening of *D. metel* extracts, 12 of these identified compounds were estimated as the major contents containing alkaloids (dihydroferuperine, hyoscine, *N*-hexadecanoylpyrrolidine, and convolamine), terpenoid (farnesyl acetone), lignan (schisanhenol B), steroid (bufotalinin), amino alcohol (sphinganine), naphthofuran (ambronide), propanolamine (moprolol), monoterpenoid lactone (loliolide), and unsaturated fatty acid (9Z,12E,15E-octadecatrienoic acid), respectively. Although, a few discovered compounds have been proven to possess wound healing properties. For instance, farnesyl acetone, found in *Prangos pabularia* [22], *Marrubium parviflorum* [23], and *Galenia africana* [24], has been reported as an antioxidant and antibacterial substance [22,23]. In addition, schisanhenol and its derivatives, from chemical synthesis [27] and medicinal plants [28,29], could potentially lower oxidative stress [28] and provide hepatoprotection [27,29]. Interestingly, we found only loliolide and 9Z,12E,15E-octadecatrienoic acid as reported in the methanolic extract of *D. metel* leaves [4]. Different extraction methods may cause different bioactive components of the same plant. Likewise, loliolide has also been found in *Lobostemon fruticosus* [30] and *Sargassum horneri* [31]. Previous studies have demonstrated a possible correlation between loliolide’s potential to heal wounds and the downregulation of oxidative stress, inflammatory cytokines, MMPs, and growth factor activation [4,30,31,32].

In this study, we found that *D. metel* extracts could scavenge DPPH and ABTS radicals. Similarly, both extracts could suppress the levels of TBARS and nitric oxide, which are makers of lipid peroxidation and inflammation. This result was consistent with previous data investigating *D. metel* methanolic extract that the leaves of *D. metel* could inhibit inflammation and also biofilm formation from bacteria [4]. Previous publications have demonstrated that reactive substances and nitric oxide are required for bacterial clearance in acute injuries [33]. Nevertheless, it was previously established that excessive ROS and inflammatory production could lead to a prolonged inflammatory stage of normal wound healing, resulting in the impairment of chronic ulcers [7,33]. Notably, persistent bacterial infiltration in patients with chronic underlying diseases, such as diabetes mellitus or renal failure, heightens inflammatory cytokines associated with poor wound healing [34].

We also found that the Dm1 extract could elevate the migration ability of fibroblasts using a scratch assay, leading to a high potential for proliferation of the wound recovery process. Previous data has indicated that growth factors like VEGF are necessary for fibroblast migration in wound sites [35]. Additionally, the regulation of Wnt/β-catenin and sonic hedgehog pathways in fibroblasts relates to the proliferative stage in the cutaneous wound healing process [14,15]. To determine the pathway that mediated the migration of fibroblasts, we investigated the expression of regulatory genes through the angiogenesis, Wnt/β-catenin, and sonic hedgehog pathways. In this present study, the Dm1 extract could significantly increase the genes, including *SHH*, *SMO*, and *GLI1*, mediated the sonic hedgehog pathway when compared to purmorphamine, the SMO receptor agonist. The induction of SMO receptors stimulates SHH signaling and then elevates the expression of the GLI1 transcriptional factor, resulting in cell proliferation and the prevention of cell apoptosis [36]. These data were consistent with the results of activation of the SHH pathway, which is prone to promoting cutaneous wound response [18,37]. Notably, Dm1 showed the upregulation of genes related to the sonic hedgehog pathway, which is crucial in the cell proliferation pathway and allowed Dm1 to demonstrate the positive effects of cell migration and the cell scratch assay.

In addition, the leaves extracts of *D. metel* could inhibit the expression of the MMP-2 protein in our gelatin zymography study. MMP-2 is one of the collagenase enzymes that influence collagen destruction. In the remodeling step, MMPs play a vital role in ECM degradation before newly constructed collagen [13]. However, the dysregulation of MMPs contributes to delayed matrix remodeling, affecting normal wound recovery [21]. The previous study has suggested that MMP-2 provides some drawbacks in lesions, such as strengthening and proliferation in wound repair [38]. The Dm extracts could down-regulate MMP-2 expression in human fibroblasts, resulting in decreased MMP-2 enzyme activity, which is the main cause of collagen degradation. Hence, the leaves of *D. metel* could promote the wound healing process by supporting the collagen matrix of wound tissue.

Altogether, farnesyl acetone and schisanhenol are two phytochemicals present in Dm extracts that may be associated with antioxidant and anti-inflammatory properties. Remarkably, loliolide, present in the leaves of *D. metel* [4], may be primarily responsible for inhibiting MMP-2 and promoting fibroblast migration along the sonic hedgehog pathway. Our findings suggest that Dm extracts accelerate wound healing by promoting the transition from the inflammatory stage to the proliferative stage and preserving the new collagen structure during tissue remodeling. Therefore, scCO_2_
*D. metel* leaves extracts could have a benefit for cutaneous lesions through the inflammatory, proliferative, and remodeling stages of the wound healing process.

## 4. Materials and Methods

### 4.1. Plant Materials and Extract Preparation

The leaves of *D. metel* were collected from the botanical garden, located in the Faculty of Pharmacy, Chiang Mai University, Chiang Mai, Thailand, during October 2022. Herbarium voucher specimens of *D. metel* (PNPRDU65005) were deposited in the Pharmaceutical and Natural Products Research and Development Unit (PNPRDU), Chiang Mai University, Chiang Mai, Thailand. Samples were cleaned and dried using the hot-air oven at 40 °C for 48 h. Dried leaves were crushed into the powder and stored at 4 °C before extraction. The scCO_2_ extracts were obtained by Spe-ed SFE-4 (Applied Separations, Allentown, PA, USA) in a laboratory scale unit according to the previous described [39,40]. The scCO_2_ extraction conditions were carried out according to our preliminary study. Briefly, the powdered leaves were mixed with 95% ethanol as a co-solvent in a ratio of 1:1. During the extraction stage, analytical grade CO_2_ (Lanna Industrial Gases, Chiang Mai, Thailand) was supplied into a sample loading cell for 20 min. The operational parameters were performed under a pressure of 300 or 500 bar with a constant temperature of 35 °C, releasing a flow rate of 3 L/min for 10 min. After scCO_2_ extraction, the rotary evaporator (Heidolph, Schwabach, Germany) was used to remove the residual ethanol. Extracts from the experimental condition of 300 and 500 bar were labeled as Dm1 and Dm2, respectively. All samples were kept at 4 °C prior to further analysis.

### 4.2. Quanlitative Analysis Using LC-QTOF-MS

The chemical compositions of scCO_2_ extracts were characterized by LC-QTOF-MS analysis based on the previous method [41] with slight modifications. The analytical method was performed using an Agilent 1290 Infinity II series equipped with a 6546 LC-QTOF instrument (Agilent Tech., Santa Clara, CA, USA). ZORBAX Eclipse Plus C18 (2.1 × 150 mm, 1.8 μm) was used for the chromatographic analysis for 26 min. Mobile phases consisted of acetonitrile (A) and HPLC-grade water with 0.1% formic acid (B). A gradient was set between A and B at 50% for 10 min, then raised B to 100% for 15 min, and kept at 100% B until the run ended. The flow rate was 0.2 mL/min with an injection volume of 10 μL. The instrument was operated in positive electrospray ionization (ESI) mode. The MS parameters were selected as 4500 V for the capillary voltage, 35 psi with 8 L/min for the nebulizer gas N_2_, and 320 °C for the dry-heater temperature. The mass range was *m*/*z* 50–1000. An online mass spectral database, ChemSpider, was used for compound identification.

### 4.3. Cell Culture

The immortalized human fibroblast cells (OUMS-36T-4F; JCRB1006.4F) were obtained from the JCRB Cell Bank (Osaka, Japan). Murine RAW 264.7 macrophage cells were purchased from the American Type Culture Collection (Rockville, MD, USA). The human fibroblast and macrophage cells were grown in Dulbecco’s Modified Eagle Medium (DMEM) (Gibco Life Technologies, Grand Island, NY, USA) supplemented with 10% fetal bovine serum (HyClone^TM^, GE Healthcare Life Sciences Laboratories, South Logan, UT, USA) and 1% penicillin/streptomycin (Capricorn Scientific GmbH, Ebsdorfergrund, Germany). All cells were maintained in an incubator (ESCO, Singapore) at 37 °C with 5% CO_2_ atmosphere.

### 4.4. Cytotoxicity Assay

The cell viability was determined based on the protein contents using the sulforhodamine B (SRB) assay [39]. Human fibroblast and RAW 264.7 cells (1 × 10^4^ cells/well) were placed in 96-well plates (Corning Costar Inc., Tewksbury, MA, USA) and incubated at 37 °C for 24 h. Cells were treated with the samples for 24, 48, and 72 h, respectively. Then, the cells were fixed to the plate using 50% trichloroacetic acid (PanReac AppliChem, Barcelona, Spain) in a refrigerator at 4 °C for 1 h. The attached cells were stained with SRB solution (Sigma Chemical, St. Louis, MO, USA) at room temperature for 30 min before being measured at 515 nm by a microplate reader (EZ 2000, Biochrom, Cambridge, UK). The percentage of cell viability was calculated using the following equation:(1)$$\mathrm{Cell}\;\mathrm{viability}\;(\%)\;=\;\frac{\left(\mathrm{OD}\;\mathrm{treated}\mathrm{cells}\;-\;\mathrm{OD}\;\mathrm{blank}\right)}{\left(\mathrm{OD}\;\mathrm{untreated}\;\mathrm{cells}\;-\;\mathrm{OD}\;\mathrm{blank}\right)}\times 100,$$

### 4.5. In Vitro Antioxidant Assay

#### 4.5.1. Scavenging Capacity Assay

The antioxidant properties of *D. metel* leaves extracts were first evaluated using DPPH and ABTS assays, according to the previous study [39]. Dm extracts were dissolved in 95% ethanol and then diluted to achieve a concentration ranging between 0.039 and 0.625 mg/mL. The sample (100 μL) was mixed with 50 μL of DPPH solution (Sigma Chemical, St. Louis, MO, USA), incubated in the dark at room temperature for 30 min, followed by measuring the radical formation at 515 nm. For the ABTS assay, the ABTS (Sigma Chemical, St. Louis, MO, USA) was first prepared in potassium persulfate in the dark at room temperature for 18 h. The stock solution was diluted with distilled water to obtain an absorbance of 0.7–0.9 units at 734 nm. The working ABTS solution (200 μL) was mixed with the sample (25 μL) at room temperature for 8 min. The absorbance was determined at 734 nm. Trolox (Sigma Chemical, St. Louis, MO, USA) was used as a standard scavenger. The IC_50_ values illustrated the fifty percent inhibitory concentration against DPPH and ABTS radicals.

#### 4.5.2. Thiobarbituric Acid-Reactive Substances (TBARS) Method

The antioxidant activity of each sample was conducted on the thiobarbituric acid-reactive substances (TBARS) assay [42]. Briefly, fibroblasts were seeded in six-well plates (2.5 × 10^5^ cells/well) and allowed to attach for 24 h. The cells were first pretreated with serum-free DMEM (control group), l-ascorbic acid (Sigma Chemical, St. Louis, MO, USA), or *D. metel* extracts for 24 h before being exposed to H_2_O_2_. All samples were tested at the same concentration of 0.00195 mg/mL. The cell lysates were collected to react with the mixture (containing 1% Triton X-100 (VWR Life Science, Solon, OH, USA), 0.6% thiobarbituric acid (BDH Chem. Ltd., Poole, UK), and 15% trichloroacetic acid), heated for 10 min, and suddenly cooled in the freezer (−20 °C) for 5 min. The final product of lipid-peroxidation was measured at 532 nm. The MDA levels in fibroblast cells were calculated in comparison to the control group.

### 4.6. Nitric Oxide Assay

The measurement of the total nitrite was assessed by the Griess reagent (Invitrogen, Thermo Fisher Scientific, Inc., Eugene, OR, USA), as previously described [16]. Nitric oxide production is represented by the concentration of nitrite in the culture medium. RAW 264.7 macrophages were placed in 96-well plates at a density of 1 × 10^4^ cells/well for 24 h. Then, the cells were pre-exposed to the serum-free medium (control group), diclofenac sodium, or *D. metel* extracts. All samples were tested at the same concentration of 0.0039 mg/mL. After incubation for 1 h, the cells were induced with 1 μg/mL lipopolysaccharides (Sigma Chemical, St. Louis, MO, USA) and incubated further for 24 h. After that, the cell supernatant (150 μL) was collected to react with 20 μL Griess reagent (a mixture of *N*-(1-naphthyl)-ethylenediamine and sulfanilic acid). The calibration curve of the standard nitrite was linear over the range of 0.78–50 μM. The absorbance of the reaction was measured at 548 nm. The nitrite concentration in the sample was calculated using the standard curve equation of nitrite (y = 0.0107x + 0.0109, *R*^2^ = 0.9991).

### 4.7. Scratch Assay

The scratch assay was used to determine the migration capacities of samples, based on the previous method [43] with some modifications. Briefly, human fibroblasts were seeded in 96-well plates at a density of 2 × 10^4^ cells/well. After 24 h of incubation, the scratch line was conducted using a 200 μL sterile pipette tip. The cell debris was washed and removed before adding serum-free DMEM (control group), the commercial deproteinized dialysate from the calf blood extract, or *D. metel* extracts. All samples were tested at the same concentration of 0.00195 mg/mL. The scratched area was imaged using an inverted microscope attached to a digital camera (Olympus E-PL3) and evaluated using Image J software (NIH, Bethesda, MD, USA) at 0, 24, 36, and 48 h after treatment. The percentage of wound closure was calculated using the following equation:(2)$$\mathrm{Wound}\;\mathrm{closure}\;(\%)\;=\;\frac{\left(\mathrm{Scratched}\;\mathrm{area}\;\mathrm{at}\;0\;h\;-\;\mathrm{Scratched}\;\mathrm{area}\;\mathrm{at}\;\mathrm{specified}\;\mathrm{time}\right)}{\mathrm{Scratched}\;\mathrm{area}\;\mathrm{at}\;0\;h}\times 100,$$

### 4.8. Gelatin Zymography for MMP-2 Activity Analysis

MMP-2 activity was evaluated by using gelatin zymography on 10% SDS-PAGE gels containing 0.1% gelatin, as previously described [42]. After treatment, the culture medium was collected and centrifuged at 3000× *g* for 3 min to remove the cell debris. After electrophoresis, gels were washed in 2.5% Triton X-100 solution for 20 min to remove SDS. The gels were then incubated for 24 h at 37 °C in the developing buffer, which contained 50 mM Tris (pH 7.5) (Vivantis, Selangor, Malaysia), 5 mM CaCl_2_, and 0.01% sodium azide (Bio Basic, Markham, Ontario, Canada). After incubation, gels were stained with 0.5% Coomassie brilliant blue R-250 (Bio Basic, Markham, ON, Canada) for 60 min and de-stained in a solution of 30% methanol and 10% glacial acetic acid (RCI Labscan, Bangkok, Thailand). The bands were detected and analyzed using the Gel Doc™ EZ System (Version 3.0; Bio-Rad, Hercules, CA, USA). The MMP-2 activated form was detected at approximately 63 kDa, based on the reference protein ladder. Relative band intensities were quantified in comparison to the control group.

### 4.9. Semi-Quantitative RT-PCR Analysis

Human fibroblasts were treated with the commercial deproteinized dialysate from the calf blood extract or Dm extracts for 48 h. Purmorphamine (Wuhan W&Z Biotech (Wuhan, China) was used as a positive control for the sonic hedgehog pathway. All samples were tested at the same concentration of 0.00195 mg/mL. Total RNA was extracted from cultured fibroblasts using the E.Z.N.A.^®^ Total RNA Kit I (Omega Bio-Tek, Norcross, GA, USA). cDNA synthesis was performed using the MyTaq™ One-Step RT-PCR Kit (Bioline, Memphis, TN, USA). Primer sequences used in this study were displayed in Table 4. The PCR products were electrophoresed on 1% agarose gel, as previously reported [16]. The bands of the agarose gel were detected using the Gel Doc™ EZ System (Version 3.0; Bio-Rad) and Image Lab™ software (Bio-Rad). The reference gene (*GAPDH*) was used to normalize the expression levels of *VEGF*, *CTNNB1*, *SHH*, *SMO*, and *GLI1*.

### 4.10. Statistical Analysis

All experiments were carried out in triplicate, and the results were expressed as mean ± standard deviation (SD) in each group. Data were calculated using one-way analysis of variance (ANOVA), followed by a LSD post hoc test (SPSS 23.0 software, Chicago, IL, USA). The significant value of *p* < 0.05 was considered statistically different.

## 5. Conclusions

The green extraction method (scCO_2_) was used to obtain the *D. metel* leaves extracts, Dm1 and Dm2. Through this study, both extracts greatly influenced the inflammatory stage of wound healing mechanisms regarding their antioxidant and anti-inflammatory effects based on the decreased levels of oxidative stress and nitric oxide. Additionally, through regulatory genes related to the sonic hedgehog pathway, Dm1 significantly improved the migrating capacity based on the fibroblast cell scratch assay, comparable with the commercial deproteinized dialysate from the calf blood extract. Considering, the improving wound site proliferation positively correlates with the fibroblasts’ ability to migrate rapidly. According to gelatin zymography, Dm1 and Dm2 could decrease the activity of MMP-2, the enzyme that breaks down collagen, which would speed up the healing process, particularly during the remodeling phase. Interestingly, farnesyl acetone, schisanhenol, and loliolide in Dm leaves extracts were identified as effective bioactive compounds for wound healing applications.

## Figures and Tables

**Figure 1 plants-12-02546-f001:**
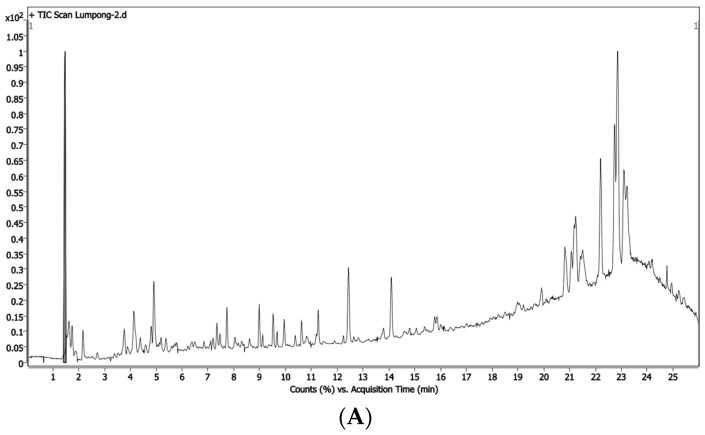
LC-QTOF-MS chromatogram of (**A**) Dm1 (black line) and (**B**) Dm2 (green line) extracts.

**Figure 2 plants-12-02546-f002:**
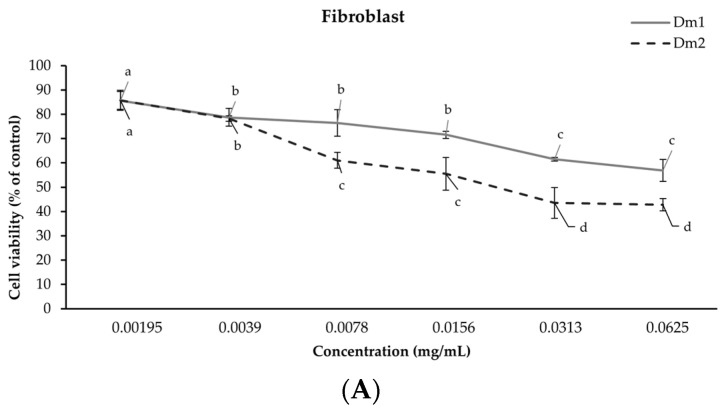
Effects of *Datura metel* extracts on cell viability after 72 h of exposure for (**A**) fibroblast; (**B**) RAW 264.7 cells using the sulforhodamine B (SRB) assay. Values were expressed as mean ± SD for triplicates in each group. Different letters (a, b, c, and d) indicated significant differences (*p <* 0.05) compared to other concentrations according to one-way analysis of variance (ANOVA), followed by a LSD post hoc test.

**Figure 3 plants-12-02546-f003:**
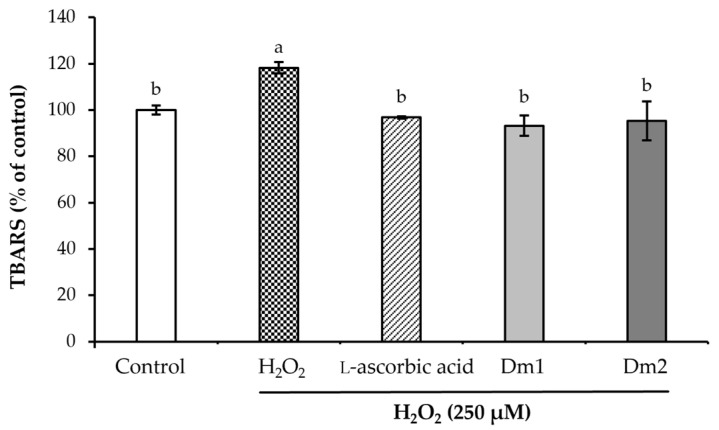
Effects of *Datura metel* extracts and l-ascorbic acid (0.00195 mg/mL) on the malondialdehyde production in hydrogen peroxide (H_2_O_2_)-induced fibroblasts using the thiobarbituric acid reactive substances (TBARS) test. Values were expressed as mean ± SD for triplicates in each group. Different letters (a and b) above the bars indicated significant differences (*p <* 0.05) compared to the control group according to one-way analysis of variance (ANOVA), followed by a LSD post hoc test.

**Figure 4 plants-12-02546-f004:**
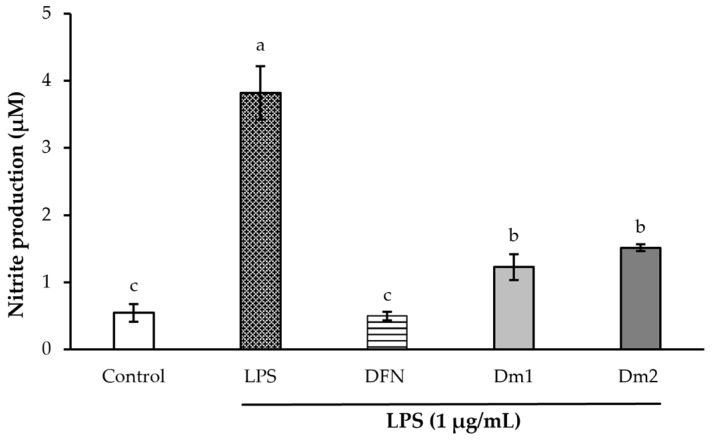
Effects of *Datura metel* extracts and diclofenac sodium (DFN) (0.0039 mg/mL) on the nitrite production in lipopolysaccharide (LPS)-induced macrophages based on the Griess reaction assay. Values were expressed as mean ± SD for triplicates in each group. Different letters (a, b, and c) above the bars indicated significant differences (*p <* 0.05) compared to the control group according to one-way analysis of variance (ANOVA), followed by an LSD post hoc test.

**Figure 5 plants-12-02546-f005:**
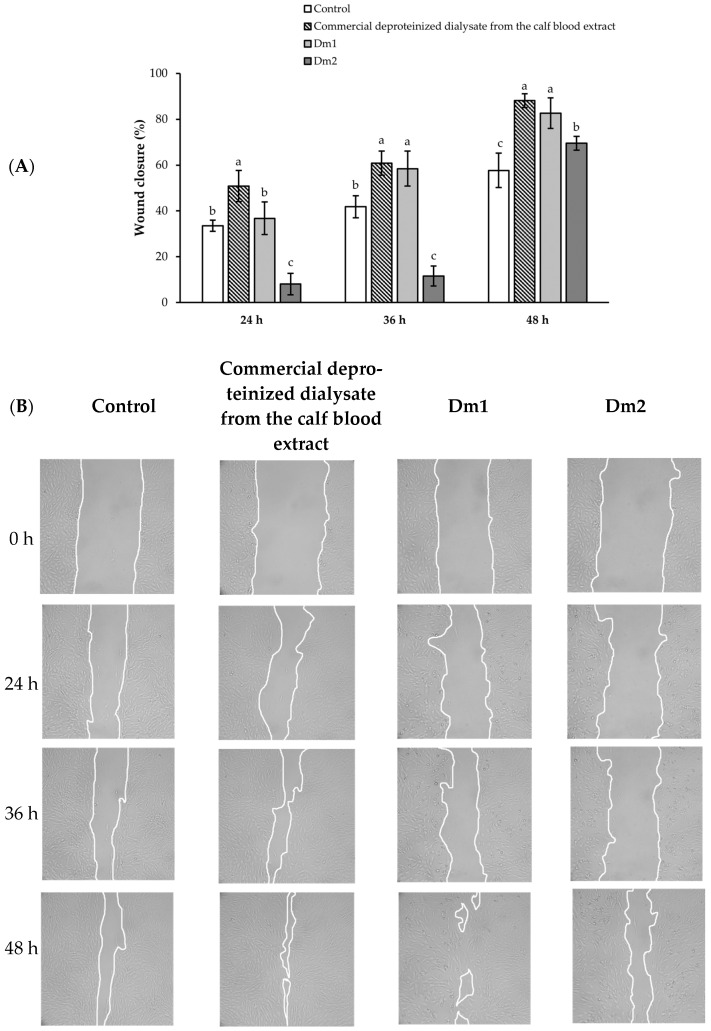
Effects of *Datura metel* extracts and the commercial deproteinized dialysate from the calf blood extract (0.00195 mg/mL) on the migration of human fibroblasts using the scratch assay (**A**) The percentage of wound closure at each time point was compared to the scratch area at 0 h; (**B**) Microscopical images represented fibroblast wound areas at 0, 24, 36, and 48 h after exposure to each sample. Values were expressed as mean ± SD for triplicates in each group. Different letters (a, b, and c) above the bars indicated significant differences (*p <* 0.05) compared to the control group at the same time point according to one-way analysis of variance (ANOVA), followed by a LSD post hoc test.

**Figure 6 plants-12-02546-f006:**
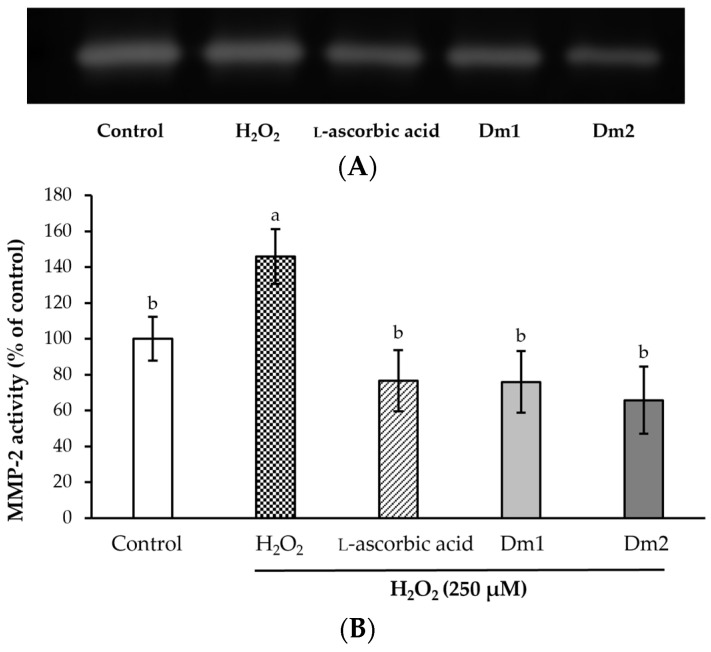
Effects of *Datura metel* extracts and l-ascorbic acid (0.00195 mg/mL) on the MMP-2 activity in hydrogen peroxide (H_2_O_2_)-induced fibroblasts using the gelatin zymography technique. (**A**) The zymogram of MMP-2 protein; (**B**) The percentage of MMP-2 activity after exposure to each sample. Values were expressed as mean ± SD for triplicates in each group. Different letters (a and b) above the bars indicated significant differences (*p <* 0.05) compared to the control group according to one-way analysis of variance (ANOVA), followed by a LSD post hoc test.

**Figure 7 plants-12-02546-f007:**
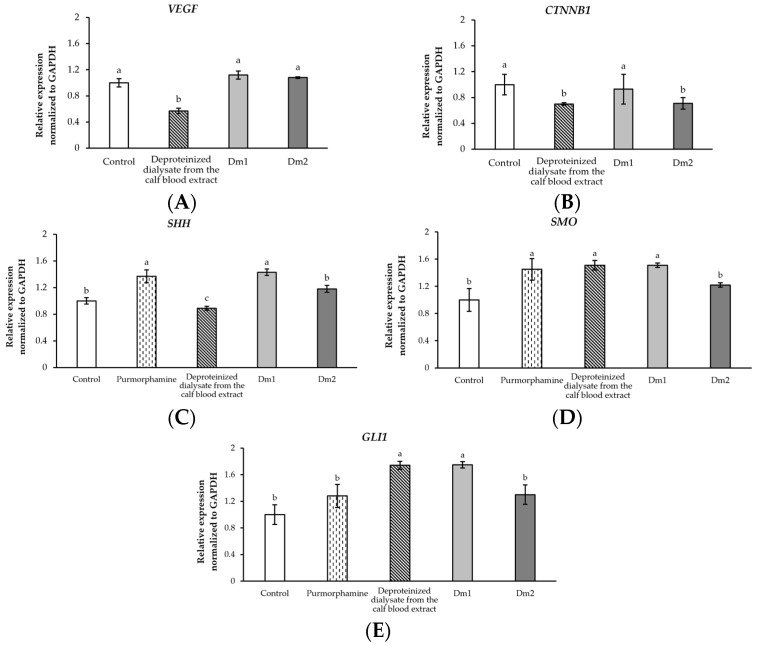
Effects of *Datura metel* extracts, the commercial deproteinized dialysate from the calf blood extract, and purmorphamine (0.00195 mg/mL) on the relative mRNA expression of genes associated with the angiogenesis (**A**) *VEGF*; Wnt/β-catenin signaling (**B**) *CTNNB1*; and sonic hedgehog pathways (**C**) *SHH*; (**D**) *SMO*; (**E**) *GLI1* in human fibroblast cells. Values were expressed as mean ± SD for triplicates in each group. Different letters (a, b, and c) above the bars indicated significant differences (*p <* 0.05) compared to the control group according to one-way analysis of variance (ANOVA), followed by a LSD post hoc test.

**Table 1 plants-12-02546-t001:** Extraction conditions and extraction yields of *Datura metel* extracts.

Experimental Conditions	Dm1	Dm2
Temperature (°C)	35	35
Pressure (bar)	300	500
Co-solvent	95% Ethanol	95% Ethanol
Yield of Extract (% *w*/*w*)	2.7 ± 0.2	3.0 ± 0.1

**Table 2 plants-12-02546-t002:** LC-QTOF-MS characterization of bioactive compounds from *Datura metel* extracts.

	Name	Molecular Formation	RT (min)	Mass (*m*/*z*)	Matching Score (%)	
					Dm1	Dm2
1	Dihydroferuperine	C_17_H_23_NO_3_	4.923	289.1681	99.48	99.08
2	Farnesyl acetone	C_18_H_30_O	20.831	262.2300	98.79	97.99
3	Schisanhenol B	C_22_H_26_O_6_	12.443	386.1730	99.47	99.48
4	Hyoscine	C_17_H_21_NO_4_	4.151	303.1472	85.42	85.27
5	Bufotalinin	C_24_H_30_O_6_	14.099	414.2043	99.62	99.63
6	*N*-Hexadecanoylpyrrolidine	C_20_H_39_NO	25.231	309.3034	99.74	99.71
7	Sphinganine	C_16_H_35_NO_2_	10.628	273.2669	99.77	99.83
8	Convolamine	C_17_H_23_NO_4_	4.213	305.1628	99.57	99.79
9	Ambronide	C_16_H_28_O	19.920	236.2145	98.82	98.65
10	Moprolol	C_13_H_21_NO_3_	4.393	239.1523	99.73	99.73
11	Loliolide	C_11_H_16_O_3_	6.964	196.1098	85.58	86.28
12	9Z,12E,15E-Octadecatrienoic acid	C_18_H_30_O_2_	21.592	278.2244	84.82	82.68

**Table 3 plants-12-02546-t003:** Antioxidative efficiency of *Datura metel* extracts.

Samples	IC_50_ for DPPH (mg/mL)	IC_50_ for ABTS (mg/mL)
Dm1	0.35 ± 0.00 ^a^	0.47 ± 0.04 ^a^
Dm2	0.31 ± 0.01 ^a^	0.36 ± 0.01 ^a^
Trolox	0.02 ± 0.00 ^b^	0.16 ± 0.00 ^b^

Values were expressed as mean ± SD for triplicates in each group. Different letters (a and b) indicated significant differences (*p <* 0.05) compared to Trolox according to one-way analysis of variance (ANOVA), followed by a LSD post hoc test.

**Table 4 plants-12-02546-t004:** Primer sequences for semi-quantitative RT-PCR.

Functional Pathway	Genes	Sequences
Angiogenesis pathway	*VEGF*	Forward: CTACCTCCACCATGCCAAGT Reverse: GCGAGTCTGTGTTTTTGCAG
Wnt/β-catenin signaling	*CTNNB1*	Forward: CCCACTAATGTCCAGCGTTT Reverse: AACCAAGCATTTTCACCAGG
Sonic hedgehog pathway	*SHH*	Forward: AAAAGCTGACCCCTTTAGCC Reverse: GCTCCGGTGTTTTCTTCATC
	*SMO*	Forward: GAAGTGCCCTTGGTTCGGACA Reverse: CCGCCAGTCAGCCACGAAT
	*GLI1*	Forward: GCAGGGAGTGCAGCCAATACAG Reverse: GAGCGGCGGCTGACAGTATA
Reference gene	*GAPDH*	Forward: GGAAGGTGAAGGTCGGAGTC Reverse: CTCAGCCTTGACGGTGCCATG

## Data Availability

The data presented in this study are available on request from the corresponding author.

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
