# Peer review of "Wound Healing Effect of Supercritical Carbon Dioxide Datura metel L. Leaves Extracts: An In Vitro Study of Anti-Inflammation, Cell Migration, MMP-2 Inhibition, and the Modulation of the Sonic Hedgehog Pathway in Human Fibroblasts"

_plants, 2023, doi:10.3390/plants12132546_

Round 1

Reviewer 1 Report

This manuscript employed the supercritical carbon dioxide (scCO2) technique in the extraction of D. metel leaves and evaluated the wound healing activity of the leaf extracts of D. metel. The work was well conceived and designed, but some necessary experimental data are lacking. Thus, this work is suitable for publication after major modifications.

(1) line 36-37: The author listed eight keywords in the article, 5-6 keywords are the most appropriate, the author should modify it. According to the contents of whole manuscript, most of keywords are not necessary, eg, antioxidants, anti-inflammation.

(2) line 132, the sentence described that “The antioxidant properties of leaf extracts were tested at concentrations ranging from 0.039 – 0.625 mg/mL”, but “Table 3” only listed experimental data for a single concentration of leaf extracts, the author should verify that.

 (3) Line 120, no data plot for 2.3 cell viability experiment, the authors should provide the data.

Author Response

This manuscript employed the supercritical carbon dioxide (scCO2) technique in the extraction of D. metel leaves and evaluated the wound healing activity of the leaf extracts of D. metel. The work was well conceived and designed, but some necessary experimental data are lacking. Thus, this work is suitable for publication after major modifications.

  1. line 36-37: The author listed eight keywords in the article, 5-6 keywords are the most appropriate, the author should modify it. According to the contents of whole manuscript, most of keywords are not necessary, eg, antioxidants, anti-inflammation.

Response: Thank you for your suggestion. ‘Antioxidant and anti-inflammation’ have been removed from keywords. Please see page 2 lines 46 – 47 of the revised manuscript.

  1. line 132, the sentence described that “The antioxidant properties of leaf extracts were tested at concentrations ranging from 0.039 – 0.625 mg/mL”, but “Table 3” only listed experimental data for a single concentration of leaf extracts, the author should verify that.

Response: Actually, we reported the antioxidant properties in Table 3 in terms of Trolox equivalent antioxidant capacity (TEAC). We assessed all samples (Dm1, Dm2, and Trolox) at concentrations ranging from 0.039 – 0.625 mg/mL in triplicate. However, in response to your comment, we have provided the antioxidant properties using IC50 values to provide clearer data. Please see Table 3, page 5 lines 148 – 152.

Table 3. Antioxidative efficiency of Datura metel extracts

Samples

IC50 for DPPH (mg/mL)

IC50 for ABTS (mg/mL)

Dm1

0.35 ± 0.00 a

0.47 ± 0.04 a

Dm2

0.31 ± 0.01 a

0.36 ± 0.01 a

Trolox

0.02 ± 0.00 b

0.16 ± 0.00 b

Values were expressed as mean ± SD for triplicates in each group. Different letters (a and b) indicated significant difference (p < 0.05) compared to Trolox according to one-way analysis of variance (ANOVA), followed by LSD post hoc test.

  1. Line 120, no data plot for 2.3 cell viability experiment, the authors should provide the data.

Response: Additional information of cell viability has been provided in Figure 2 within the revised manuscript. Please see pages 4-5 lines 134 – 138. 

  1. Quality of English Language: I am not qualified to assess the quality of English in this paper.

Response: Thank you for your valuable comments. The revised manuscript has been thoroughly checked and edited according to the grammar issue using Grammarly Premium and Quill Bot program.

Reviewer 2 Report

The authors have prepared Dm1 and Dm2 extracts from the leaves of the thorn apple, using scCO2 procedures at different pressures of 300 and 500 bar, respectively, and studied properties of these extracts in in vitro assays. In general, the range of tests and methods used are good and research has been in-depth, covering signalling pathways as well. However, some inaccuracies in the presentation of results should be corrected.

Minor:

1.         It is accepted as a role to use International Non-Proprietary Name (INN) in the scientific papers. Is it impossible here to use INN for Solcoseryl®?  

2.         Explain unusual concentration diapason such as each extract at a concentration ranging from 0.00195 to 0.0625 mg/mL. What was reason to choose exactly these concentrations?

3.         How to explain extracts effect differences in the wound healing test if contents of Dm1 and Dm2 are almost identical?

4.         In Methods description of concentrations of the substances used in the assays is missing. The same information is missing in the legends of figures.

5.         In vitro assays drug concentrations often do not correlate with effects on the cell viability. Moreover, if the authors showed highest concentrations for each cell line viability such as 0.00195 mg/mL for human fibroblasts and 0.0039 mg/mL for RAW 264.7 cells, why then standard  L-ascorbic acid, and diclofenac sodium the same concentration (0.0039 mg/mL) was chosen for fibroblasts and macrophages?

6.         Fig.5A representative picture of the band intensities of the level of active MMP-2 in fibroblast cells does not show increase of the H2O2 treatment. How was 1.5 increase calculated?

7.         It would be more understandable if statistical differences would be explained versus what they are calculated? The explanation “Different letters (a, b, and c) above the bars indicated significant differences (p < 0.05) within treatments” do not show when comparison is done versus control, when versus standard, when versus other concentrations etc.

Author Response

The authors have prepared Dm1 and Dm2 extracts from the leaves of the thorn apple, using scCO2 procedures at different pressures of 300 and 500 bar, respectively, and studied properties of these extracts in in vitro assays. In general, the range of tests and methods used are good and research has been in-depth, covering signalling pathways as well. However, some inaccuracies in the presentation of results should be corrected.

  1. It is accepted as a role to use International Non-Proprietary Name (INN) in the scientific papers. Is it impossible here to use INN for Solcoseryl®?

Response: Thank you for your comment. The INN name of Solcoseryl®, “deproteinized dialysate from the calf blood extract” has been used throughout the revised manuscript.

  1. Explain unusual concentration diapason such as each extract at a concentration ranging from 0.00195 to 0.0625 mg/mL. What was reason to choose exactly these concentrations?

Response: In the screening of antioxidant property, we initially tested the extracts at the concentrations ranging from 0.039 – 0.625 mg/mL.  Subsequently, we performed using 10-times lower concentrations in the cell viability assay. On the other hand, we found that the maximum non-cytotoxic concentration (considered at 80 % cell viability [1]) for fibroblast cells was 0.00195 mg/mL, and for macrophage cells, it was 0.0039 mg/mL. Therefore, the maximum concentration that gave 80% cell viability was selected to use in further cell-based experiments.  

Reference:

  1. Ruksiriwanich, W.; Khantham, C.; Sringarm, K.; Sommano, S.; Jantrawut, P. Depigmented Centella asiatica extraction by pretreated with supercritical carbon dioxide fluid for wound healing application. Processes 2020, 8, 277.

  1. How to explain extracts effect differences in the wound healing test if contents of Dm1 and Dm2 are almost identical?

Response: As mentioned in the discussion section, we performed various extraction methods for D. metel in our preliminary studies. Through this exploration, we discovered that the crude extracts from scCO2 with 95% ethanol as a co-solvent promising potential for wound healing research. These findings suggest that the specific extraction method can influence the composition and properties of the extracted compounds. Even though the contents of Dm1 and Dm2 extracts may appear almost identical, the different extraction techniques may lead to variations in the presence or concentration of certain bioactive components. These variances in composition could explain the observed differences in their effects on wound healing, specifically regarding the migration ability and signaling pathways in human fibroblasts. We will elucidate and focus on the pure active compound obtained from scCO2 extracts of D. metel in the further study.

  1. In Methods description of concentrations of the substances used in the assays is missing. The same information is missing in the legends of figures.

Response: Additional information of concentration of each substance has been added to the methods and the figure description as below:

  • Thiobarbituric Acid-Reactive Substances (TBARS) Method: “All samples were tested at the same concentration of 0.00195 mg/mL.” has been added in lines 415-416, page 13.
  • Nitric Oxide Assay: “All samples were tested at the same concentration of 0.0039 mg/mL.” has been added in lines 428-429, page 13.
  • Scratch Assay: “All samples were tested at the same concentration of 0.00195 mg/mL.” has been added in lines 422-443, page 14.
  • Semi-Quantitative RT-PCR Analysis: “All samples were tested at the same concentration of 0.00195 mg/mL.” has been added in line 464, page 14.
  • Figure 3: “Effects of Datura metel extracts and L-ascorbic acid (0.00195 mg/mL) on the malondialdehyde production”. Please see page 6 lines 162-163.
  • Figure 4: “Effects of Datura metel extracts and diclofenac sodium (DFN) (0.0039 mg/mL) on the nitrite pro-duction in lipopolysaccharide (LPS)-induced macrophages”. Please see page 6 lines 177-178.
  • Figure 5: “Effects of Datura metel extracts and deproteinized dialysate from the calf blood extract (0.00195 mg/mL) on the migration of human fibroblasts”. Please see page 8 lines 196 – 1
  • Figure 6: “Effects of Datura metel extracts and L-ascorbic acid (0.00195 mg/mL) on the MMP-2 activity in hydrogen peroxide (H2O2)-induced fibroblasts”. Please see page 8 lines 213 –
  • Figure 7: “Effects of Datura metel extracts, deproteinized dialysate from the calf blood extract, and purmorphamine (0.00195 mg/mL) on the relative mRNA expression of genes…”. Please see page 10 lines 241 –

  1. In vitro assays drug concentrations often do not correlate with effects on the cell viability. Moreover, if the authors showed highest concentrations for each cell line viability such as 0.00195 mg/mL for human fibroblasts and 0.0039 mg/mL for RAW 264.7 cells, why then standard L-ascorbic acid, and diclofenac sodium the same concentration (0.0039 mg/mL) was chosen for fibroblasts and macrophages?

Response: We are sorry for the mistake. These statements have been corrected to be “The Dm1 and Dm2 extracts were non-toxic (> 80% cell viability) in human fibroblasts and RAW 264.7 cells at concentrations of 0.00195 and 0.0039 mg/mL, respectively, after 72 h of incubation, as shown in Figure 2. The higher concentrations contributed to cytotoxic effects on the cells. Thus, the maximum non-toxic concentration for each cell line would be used in subsequent experiments.”. Please see page 4 lines 123 – 127 of the revised manuscript. Fibroblast cells were treated with Dm extracts, L-ascorbic acid, purmorphamine, and deproteinized dialysate from the calf blood extract at the same concentration of 0.00195 mg/mL. In case of macrophage (RAW 264.7), the cells were treated with Dm extracts and diclofenac sodium at the same concentration of 0.0039 mg/mL, as shown in the manuscript. The tested concentration of each sample for all tests has been corrected. In the results of cellular antioxidant activity, the concentration of L-ascorbic acid has been edited to “0.00195 mg/mL”. Please see page 5 line 157.

  1. 5A representative picture of the band intensities of the level of active MMP-2 in fibroblast cells does not show increase of the H2O2 treatment. How was 1.5 increase calculated?

Response: In the gelatin zymography for MMP-2 activity analysis, we used the fresh medium from different treated wells. The band intensities from different gels (4 replicates) were obtained from the Image Lab™ software (Bio-Rad), and the bands from each gel were captured using the Gel Doc™ EZ System. The color of the background varied upon auto-adjustment in the Gel Doc™ EZ System. The values of average H2O2 treatment were calculated as below:

No.

Figure (Left: control, Right: H2O2)

Control

(Band intensity)

H2O2 treatment

(Band intensity)

MMP activity (% of control)

1

86450

145110

167.85

2

157002

223295

142.22

3

19497

26934

138.14

4

158117

213778

135.20

Average

145.85

 **Please see the attachment.

  1. It would be more understandable if statistical differences would be explained versus what they are calculated? The explanation “Different letters (a, b, and c) above the bars indicated significant differences (p < 0.05) within treatments” do not show when comparison is done versus control, when versus standard, when versus other concentrations etc.

Response: To clarify, the description of statistical differences has been revised to “Different letters (a, b, and c) above the bars indicated significant differences (p < 0.05) compared to the control group according to one-way analysis of variance (ANOVA), followed by LSD post hoc test.” Please see below Table 3, Figure 2 to 7.

  • In Figure 2, the phrase: “Different letters (a, b, c, and d) indicated significant differences (p < 0.05) compared to other concentrations…” has been added. Please see page 5 lines 136 – 1
  • In Table 3, the phrase: “Different letters (a and b) indicated significant differences (p < 0.05) compared to the Trolox…” has been inserted. Please see page 5 lines 149 – 1
  • In Figure 3, the phrase: “Different letters (a and b) above the bars indicated significant differences (p < 0.05) compared to the control group…” has been added. Please see page 6 lines 165 – 1
  • In Figure 4, the statement: “Different letters (a, b, and c) above the bars indicated significant differences (p < 0.05) compared to the control group…” has been added. Please see page 6 lines 179 – 1
  • In Figure 5, the statement: “Different letters (a, b, and c) above the bars indicated significant differences (p < 0.05) compared to the control group…” has been appended. Please see page 8 lines 201 – 202.
  • In Figure 6, the phrase: “Different letters (a and b) above the bars indicated significant differences (p < 0.05) compared to the control group…” has been added. Please see page 8 lines 215 – 216.
  • In Figure 7, the phrase: “Different letters (a, b, and c) above the bars indicated significant differences (p < 0.05) compared to the control group…” has been inserted. Please see page 10 lines 245 –

  1. Quality of English Language: I am not qualified to assess the quality of English in this paper.

Response: Thank you for your valuable comments. The revised manuscript has been thoroughly checked and edited according to the grammar issue using Grammarly Premium and Quill Bot program.

Round 2

Reviewer 1 Report

ok

Author Response

Thank you for your thoughtful review of this manuscript.

Sincerely,

Ruksiriwanich et al.